# Nucleic acid mediated activation of a short prokaryotic Argonaute immune system

Jithesh Kottur [1,2,3] ✉, Radhika Malik [1,3] ✉ & Aneel K. Aggarwal [1] ✉

A short prokaryotic Argonaute (pAgo) TIR-APAZ (SPARTA) defense system, activated by invading DNA to unleash its TIR domain for NAD(P)$^+$ hydrolysis, was recently identified in bacteria. We report the crystal structure of SPARTA heterodimer in the absence of guide-RNA/target-ssDNA (2.66 Å) and a cryo-EM structure of the SPARTA oligomer (tetramer of heterodimers) bound to guide-RNA/target-ssDNA at nominal 3.15–3.35 Å resolution. The crystal structure provides a high-resolution view of SPARTA, revealing the APAZ domain as equivalent to the N, L1, and L2 regions of long pAgos and the MID domain containing a unique insertion (insert57). Cryo-EM structure reveals regions of the PIWI (loop10-9) and APAZ (helix αN) domains that reconfigure for nucleic-acid binding and decrypts regions/residues that reorganize to expose a positively charged pocket for higher-order assembly. The TIR domains amass in a parallel-strands arrangement for catalysis. We visualize SPARTA before and after RNA/ssDNA binding and uncover the basis of its active assembly leading to abortive infection.

There has been intense interest over the past few years in identifying new bacterial defense systems, both for their own biological interest and as potential tools for biotechnological applications[1,2]. One new defense system is the short prokaryotic Argonaute TIR-APAZ (SPARTA), which is activated by foreign (plasmid) DNA to unleash its TIR domain for NAD(P)$^+$ hydrolysis[3]. The resulting depletion of the essential metabolite NAD(P)$^+$ lends to the killing of the infected cell (abortive infection) as a means to prevent the spread of the invading DNA to the remaining bacterial cell population.

SPARTA adds to the growing list of prokaryotic Argonaute (pAgo) proteins that comprise defense systems in bacteria, with similarities to the RNA interference (RNAi) systems in eukaryotes[4–7]. The proteins use single-stranded RNA or DNA as guides to target complementary nucleic acids and can be divided into two broad classes: long pAgos and short pAgos. The long pAgos contain the same four domains and two ordered linker (L) composition (N-L1-PAZ-L2-MID-PIWI) as eukaryotic Agos, whereas the short pAgos contain only the MID (middle) and PIWI (P element-induced wimpy testis) domains[5,6]. Short pAgos, however, are associated with proteins containing an analogous to PAZ (APAZ) domain, the N-terminus of which is often fused to Toll/

interleukin-1 receptor (TIR), Silent informator regulator 2 (SIR2), or a nuclease-type domain[8]. The APAZ-containing proteins can be stand-alone and in the same operon as short pAgo or can be fused to the N-terminus of pAgo in a single polypeptide[8]. Although the TIR domain was originally described as a protein-protein interaction module in innate immune signaling in animals, recent studies have revealed TIR domains that possess enzymatic activities to metabolize NAD$^+$ in neurodegenerative disease in humans as well as defense systems in plants and bacteria[9–12].

The SPARTA system was first discovered in 2018[6] and further characterized by Koopal et al[3] in 2022, who showed in an elegant set of studies that the short p-Ago of SPARTA associates with a TIR-APAZ protein to form a catalytically inactive heterodimer with 1:1 stoichiometry. However, upon guide (g) RNA and target (t) ssDNA binding, four heterodimers transition to an oligomer (a tetramer of heterodimers) that unleashes the NAD(P)ase activity of the TIR domain for abortive infection[3].

Here, we present a high-resolution crystal structure of the SPARTA heterodimer in absence of gRNA/tDNA (Apo SPARTA) at 2.66 Å resolution and a cryo-EM structure of the SPARTA oligomer

[1]Department of Pharmacological Sciences, Icahn School of Medicine at Mount Sinai, New York, NY 10029, USA. [2]Department of Antiviral Drug Research, Institute of Advanced Virology, Thiruvananthapuram, Kerala 695317, India. [3]These authors contributed equally: Jithesh Kottur, Radhika Malik. ✉e-mail: jitheshkottur@iav.res.in; radhika.malik@mssm.edu; aneel.aggarwal@mssm.edu

bound to gRNA/tDNA at a nominal 3.15-3.35 Å resolution. Together, the crystal and cryo-EM structures provide a visualization of SPARTA before and after RNA/ssDNA binding and reveal the basis of SPARTA's active assembly leading to NAD(P)⁺ degradation and cell death.

## Results and Discussion
### Biophysical and structural analysis of SPARTA
We first co-expressed and purified the heterodimeric TIR-APAZ/short pAgo complex from *Crenotalea thermophila* (SPARTA) (Fig. 1a, b), and then used mass photometry to characterize the oligomeric state of SPARTA in the absence and presence of gRNA/tDNA. The mass distribution revealed that Apo SPARTA exists as a heterodimer with an estimated molecular weight (MW) of 109 kDa (close to the theoretical MW of 112 kDa) (Fig. 1c). Upon incubation of SPARTA with the gRNA/tDNA at 55 °C for 12 hours, we observed mass distributions corresponding to monomeric, dimeric, and tetrameric complexes of the heterodimer (Fig. 1c).

To see what the structure of SPARTA looks like in absence of gRNA/tDNA, we obtained crystals of Apo SPARTA that diffracted to 2.66 Å resolution and belonged to space group P6₅22 with unit cell dimensions a = b = 197.18 Å, c = 183.4 Å, α = β = 90°, γ = 120° with one SPARTA heterodimer in the crystallographic asymmetric unit. We solved the structure by molecular replacement using an AlphaFold[13] model of the heterodimer, followed by manual building (Fig. 1d, Supplementary Fig. 1 and Supplementary Table 1). To see how SPARTA transitions to a catalytically active oligomer, we incubated the heterodimer with gRNA/tDNA and started with single-particle EM studies of negatively stained samples. The images yielded 2D class averages showing a clear butterfly-like architecture (SPARTA oligomer) as well as smaller particles (Supplementary Fig. 2). The butterfly-like particles (SPARTA oligomer) from the negative stained sample were used as a template for further cryo-EM studies. The initial untilted cryo-EM dataset resulted in an anisotropic map, however, upon tilting (to 30° and 45°) the sphericity of the map improved substantially and resulted in an isotropic reconstruction at a nominal resolution of 3.35 Å (Supplementary Fig. 3). The reconstruction revealed four SPARTA heterodimers (A, B, C and D) assembled into a tetramer of heterodimers, with the MID, PIWI, APAZ domains and gRNA/tDNA on the outside (wings of the butterfly) and the TIR domains on the inside (body of the butterfly). The resolution of Ago-APAZ-gRNA/tDNA section was further improved by performing focused C2 symmetry-based refinements of each of the "wings" to nominal resolutions of 3.15 Å and 3.17 Å, respectively (Supplementary Fig. 4 and Supplementary Table 2). A final composite map from the individually refined components was generated and the model coordinates were refined against it (Supplementary Figs. 4, 5).

### Architecture of catalytically inactive Apo SPARTA heterodimer
The crystal structure of Apo SPARTA heterodimer reveals an extended TIR-APAZ protein interacting extensively with the more compactly folded short pAgo (MID-PIWI) protein (Fig. 1d). The TIR (residues 1 to 138) and APAZ (residues 161 to 420) domains connect via an extended linker (residues 139 to 160) and subtend a concave surface against which the MID (residues 1 to 269) and PIWI (residues 270 to 507) domains dock (Fig. 1d). The TIR domain makes contacts exclusively with the MID domain, whereas the APAZ domain makes contacts mostly with the PIWI domain and few contacts with the MID domain (Fig. 1d). This interface between TIR-APAZ and short pAgo is extensive (burying ~ 2240 Å²) and underlies the stability of the Apo SPARTA heterodimer in solution (Fig. 1d).

The SPARTA TIR domain is similar in structure to the TIR domain in proteins with scaffolding and enzymatic roles, composed of a central β-sheet (βA to βD) surrounded by helices (αA to αE) and loops (Fig. 1e). A prominent loop is the BB loop (between βB and αB) that is frequently involved in TIR self-association[11,12]. The characteristic glutamate (Glu77) in TIRs with NADase activity is located on helix αC

(Fig. 1e). The SPARTA TIR aligns with rmsds of 1.06 Å (113 Cαs), 1.05 Å (120 Cαs), 0.86 Å (119 Cαs) and 1.07 Å (126 Cαs) with the TIRs of human TLR2[14] (PDB: 1FYW), human SARM1[15] (PDB: 7NAK), plant NLR-RPP1[16] (PDB: 7DFV), and bacterial TIR-SAVED[17] (PDB: 7QQK), respectively (Fig. 1e and Supplementary Fig. 6a). Although the APAZ domain is often associated with short pAgos, its 3D structure has remained largely uncharacterized. It was originally predicted as analogous to the PAZ domain of long pAgos[7], though more recent predictions have suggested homology to the N and L1 regions of long pAgos[18,19]. The APAZ domain emerges from our structure to be topologically similar to the N, L1 and L2 regions of long pAgos and, by analogy, we divide it into APAZ-N (residues 167-280), APAZ-L1 (residues 281-352) and APAZ-L2 (residues 353-450) regions (Figs. 1a, 1f and Supplementary Fig. 6b). APAZ-N is composed of a small four-stranded β-sheet (βH-βK) flanked by three short α-helices (αG to αJ) and one long α-helix (αK) that connects to APAZ-L1. APAZ-N is linked to the TIR domain via two short β-strands (βF and βG) connected to the connector helix (αF). APAZ-L1 is dominated by two long β-strands (βO and βP) in the same manner as the L1 region of long pAgos (Fig. 1f and Supplementary Fig. 6b). The major topological difference between APAZ and long pAgos is omission of the region corresponding to the PAZ domain. In long p-Agos, L1 continues into the PAZ domain, whereas in APAZ, APAZ-L1 continues directly into APAZ-L2. APAZ-L2 is dominated by two α-helices (αL and αM) analogous to those observed in the L2 region of long pAgos. However, whereas the L2 of long pAgos continues to the MID domain, APAZ-L2 terminates in a short C-terminal helix (αN, a key element in blocking the binding of target DNA, discussed below) (Fig. 1f and Supplementary Fig. 6b). Overall, APAZ appears to be evolutionarily related to the PAZ lobe of long pAgos. However, the APAZ has lost the PAZ domain but retained regions corresponding to the N, L1 and L2 and acquired another function through fusion with an enzymatic domain such as TIR, SIR2 or a nuclease.

The SPARTA short pAgo MID domain is composed of a parallel four-stranded β-sheet (β3 to β6) surrounded by α-helices in a typical Rossmann fold (Fig. 1g). The MID domain is as expected similar in structure to other MID domains superimposing, for example, with the MID domain of long CbAgo[20] (PDB: 6QZK) with a rmsd of 5.23 Å (for 87 Cαs) (Fig. 1g and Supplementary Fig. 6c). A unique feature of the SPARTA MID domain is "insert57" found between α5 and α7, composed of a long helix (α6) and two short helices, that extends outwards from the body of the MID domain. Intriguingly, from sequence alignments, insert57 emerges as a strongly conserved feature of short pAgos but not of long pAgos (Supplementary Fig. 7). The PIWI domain is composed of a curved eight stranded β-sheet (β7 to β14) flanked by α-helices from both sides (Fig. 1g). It has an overall fold of RNAase H domain and aligns with the PIWI domain of long CbAgo with an rmsd of 3.09 Å (for 140 Cαs) (Fig. 1g and Supplementary Fig. 6c). Unlike most long pAgos, the PIWI domains of short pAgos are catalytically inactive and lack the acidic residues for the hydrolysis of DNA or RNA. In the SPARTA PIWI domain, residues Val284, Tyr328, Arg362 and Asn484 substitute for residues Asp541, Glu577, Asp611, and Asp727 in the PIWI domain of catalytically active CbAgo[20]. The SPARTA MID and PIWI domains are positioned in a configuration similar to that in other long and short pAgos, with an extensive interface, burying ~1600 Å² of surface area. Interestingly, when the MID and PIWI domains of the SPARTA heterodimer are superimposed on the MID and PIWI domains of CbAgo[20], the APAZ domain occupies the spatial volume as the N, L1 and L2 regions of CbAgo (Fig. 1g and Supplementary Fig. 6c). This lends further credence to the idea that the short pAgo defense systems may have arisen from long pAgos by losing the PAZ domain but gaining a TIR, SIR2 or a nuclease domain instead.

### Architecture of catalytically active SPARTA oligomer
The cryo-EM structure of SPARTA in the presence of the gRNA/tDNA reveals four SPARTA heterodimers (A, B, C and D) assembled into a

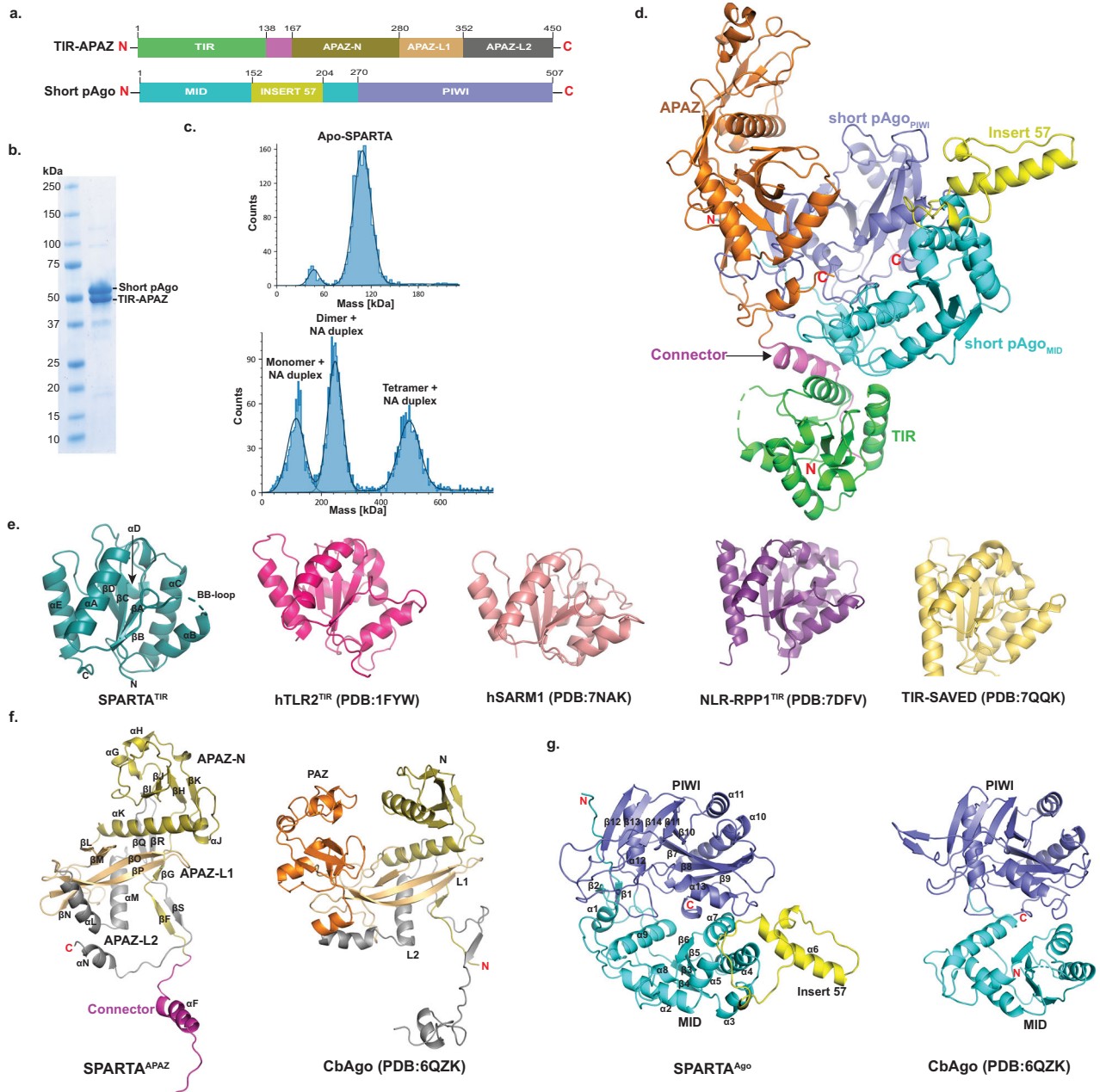

**Fig. 1 | Overall structure of Apo SPARTA. a** Schematic diagram of the domain architecture of TIR-APAZ and short pAgo. **b** SDS-PAGE of co-purified short pAgo (58.2 kDa) and TIR-APAZ (53.2 kDa) proteins. Uncropped gel image is provided as a Source Data file. The purification of the SPARTA heterodimer was repeated at least 5 times with similar results. **c** Mass photometry data of the Apo SPARTA (top) and SPARTA incubated with guide RNA (gRNA) and target DNA (tDNA) heteroduplex (bottom). Apo SPARTA shows a clear homogeneous population for the monomer. Upon incubation with gRNA/tDNA, SPARTA additionally shows the formation of dimer and tetramer complexes. **d** Crystal structure of TIR-APAZ/short pAgo heterodimeric complex. TIR domain, connector and APAZ domain are colored as green, magenta and orange, respectively. MID, insert57 and PIWI domains of short

pAgo are colored as cyan, yellow and slate, respectively. **e** Structural comparison of monomeric TIR domains from TIR domain in proteins with scaffolding and enzymatic roles. The secondary structure elements are labeled for SPARTA TIR. **f** Structural comparison of SPARTA APAZ domain with the N-PAZ-L1-L2 regions of the *Clostridium butyricum* (Cb) long pAgo (PDB ID: 6QZK). The N domain, L1, PAZ, and L2 of CbAgo are colored as olive, orange, light orange and grey, respectively. Based on topological similarity, the APAZ-N, APAZ-L1 and APAZ-L2 of SPARTA APAZ are colored similarly to the CbAgo. **g** Structural comparison of SPARTA short pAgo with MID and PIWI domains of the Cb long pAgo (PDB ID: 6QZK). The domains are similar in structure with the exception of the presence of Insert57 in SPARTA short pAgo.

tetramer of heterodimers (Fig. 2a, b). The oligomer resembles a butterfly, with the MID, PIWI and APAZ domains on the outside ("wings') and the TIR domains on the inside (the "body") (Fig. 2a, b). The symmetry of the oligomer is unusual. The MID, PIWI and APAZ domains of heterodimers A and B (or C and D) are related by two-fold symmetry but $TIR_A$ and $TIR_B$ (or $TIR_C$ and $TIR_D$) are related to each other by translational symmetry (due to ~175° rotation of $TIR_B$- described below)

(Fig. 2). Further, $TIR_A$-$TIR_B$ are related to $TIR_C$-$TIR_D$ by screw-like symmetry resulting in a TIR assembly akin to the "parallel-strands" arrangement observed in scaffolding proteins (Myd88 and MAL, for example) as opposed to the "antiparallel-strands" arrangement of TIRs in enzymatic proteins (SARM1 and RPP1, for example)[21] (Fig. 2c). However, the enzymatic TIRs of AbTir (from *Acinetobacter baumannii*) have also recently been found to assemble through the parallel-strands

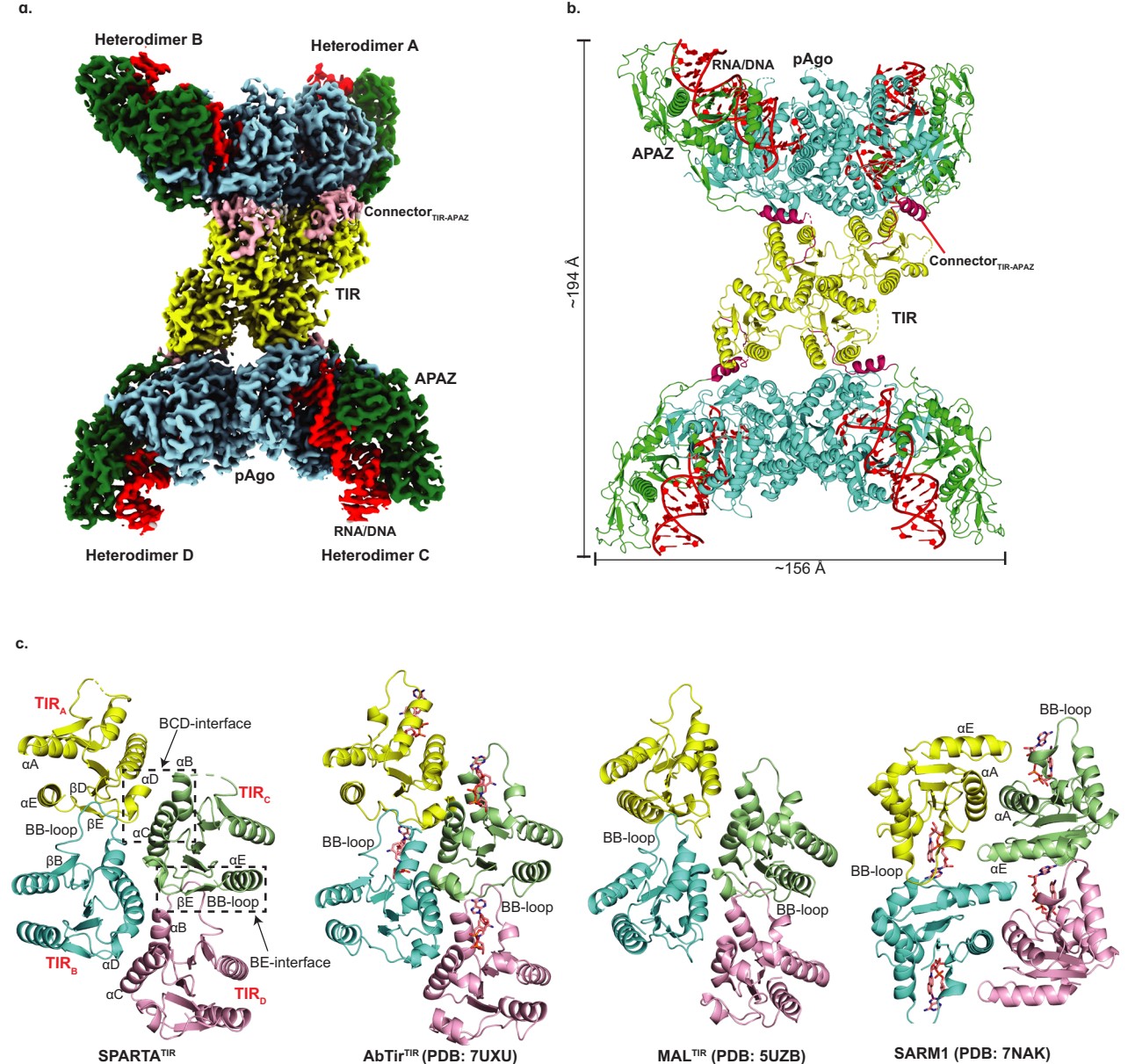

**Fig. 2 | Architecture of the catalytically active SPARTA oligomer. a** The cryo-EM map shows the domain organization of the "butterfly" shaped SPARTA oligomer with gRNA/tDNA. The oligomer is organized as a tetramer of heterodimers (labeled as A, B, C and D). The TIR, APAZ, short pAgo (MID-PIWI) are shown in yellow, green and blue, respectively. The connector between the TIR and APAZ domain is shown in pink and the gRNA/tDNA is depicted in red. This composite cryo-EM map was obtained by using C2 based focused symmetry refinement of both 'wings' and local refinement of TIR domains. **b** Structural arrangement of the SPARTA oligomer. The protein and nucleic acid components are colored as in **a**. Dimensions of the complex are given in Å. **c** Structural comparison of the tetrameric arrangement of TIR domain assembly in SPARTA oligomer versus that in select proteins with scaffolding (MAL) and enzymatic (AbTir and SARM1) roles. The secondary structure elements are labeled for SPARTA TIR. AbTir and SARM1 are depicted with bound NAD[+] analogue (shown in sticks).

arrangement[22] (described below) (Fig. 2c), suggesting versatility in how enzymatic TIRs can organize for catalysis.

### Interactions with guide RNA and target DNA

Each SPARTA heterodimer binds a single gRNA/tDNA duplex via their MID, PIWI and APAZ domains (Figs. 2a, b, and 3). The guide RNA can be traced for a segment corresponding to nucleotides 1-19 and the target DNA to nucleotides 2'-19'. The 3-19 segment of guide RNA is base-paired to the 3'–19' segment of the target DNA and the conformation of the helix is A-form (Fig. 3). Many of the RNA:DNA binding features mimic those observed with long pAgos[5,23]. The 5' phosphate of the guide strand, for example, inserts into a binding pocket in the MID domain and the first base (U1) is splayed out of the helix (Fig. 3a). The 5'

phosphate is held by hydrogen bonds with His207 and electrostatic interactions with Lys211 of the MID domain. In addition, a Mg[2+] ion is observed in the MID binding pocket that is coordinated to the 5' phosphate, A3 phosphate, and Asn468 (and putatively to the main chain carbonyl of Ile507) (Fig. 3a). Together, the observed interactions to the 5' phosphate help to explain requirement for a 5' phosphate on the guide RNA. U1 stacks against His207 and makes a direct hydrogen bond with Tyr148 of the MID domain, reminiscent of the interactions observed with long pAgos[5,23].

The gRNA/tDNA duplex as a whole is accommodated in a channel that runs between the MID-PIWI and the APAZ domains. Most of the interactions are to the sugar-phosphate backbones of the gRNA and tDNA, though there are some contacts to the bases (Fig. 3).

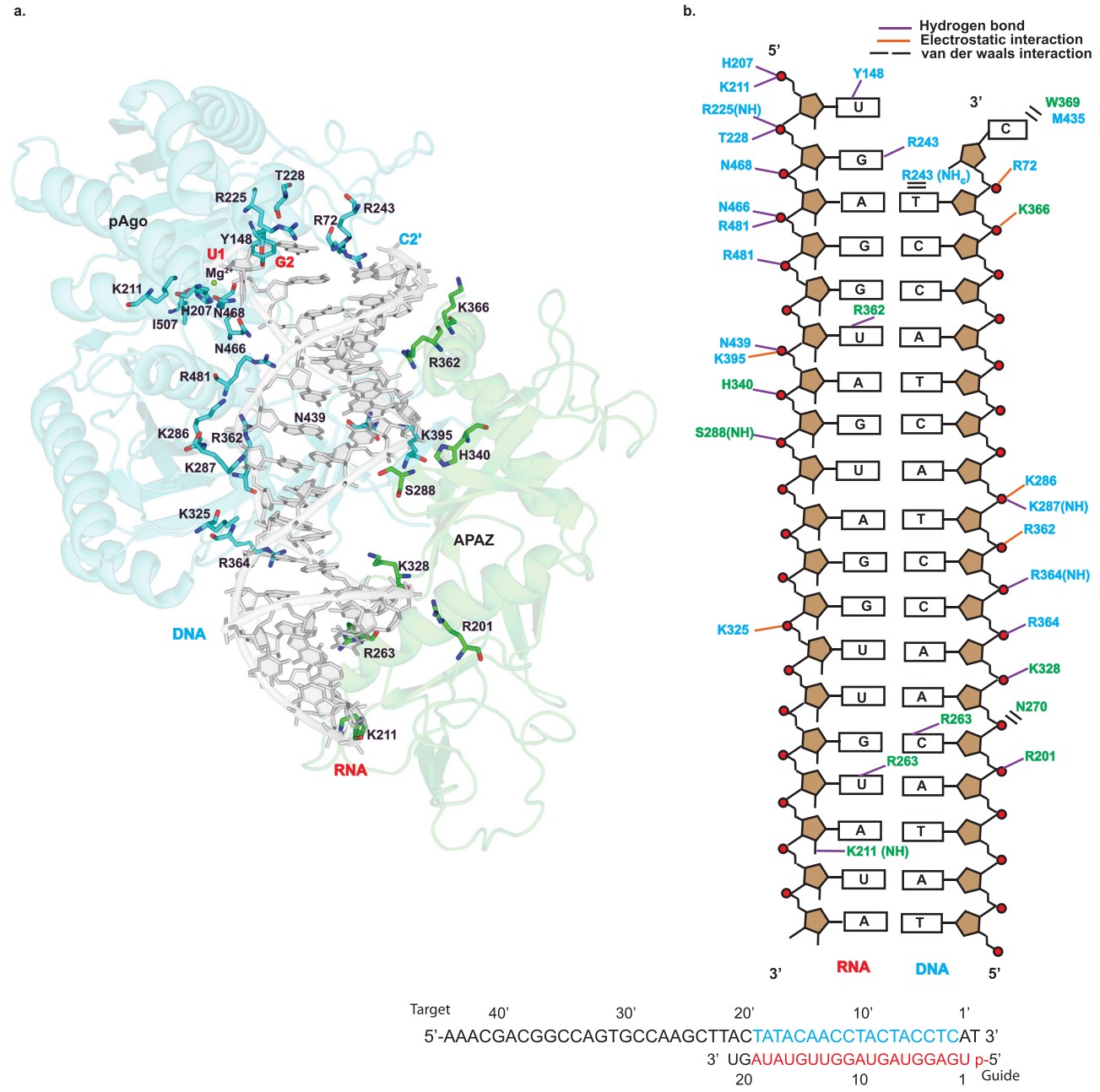

**Fig. 3 | MID-PIWI and the APAZ domain grip the gRNA/tDNA in the catalytically active SPARTA oligomer. a** The gRNA/tDNA duplex as a whole is accommodated in a channel that runs between the MID-PIWI and the APAZ domains. DNA is shown in light grey and the cartoon representation of MID-PIWI and APAZ domains are shown in blue and green, respectively. Key residues interacting with both the gRNA and tDNA strands as well as Mg²⁺ ion are highlighted. **b** Schematic representation of the interactions of MID-PIWI and APAZ domains with gRNA/tDNA strands. Amino acid residues of the MID-PIWI and APAZ domain are indicated by blue and green colors, respectively. DNA backbone phosphates are shown as red circles. Hydrogen bonds are indicated by purple lines (determined by interaction distance <3.6 Å), electrostatic interactions are shown by orange lines and van der waals interactions are shown by black bars. All the bases constituting the gRNA and tDNA strands used for the reconstitution of the SPARTA complex are shown at the bottom. The modeled RNA, DNA and unmodeled bases in the catalytically active SPARTA oligomer are shown in red, blue and black letters, respectively.

Hydrogen bonds and/or electrostatic interactions to the sugar-phosphate backbones are from residues Arg72, His207, Lys211, Arg225, Thr228 of the MID domain, Lys286, Lys287, Lys325, Arg362, Arg364, Lys395, Asn439, Asn466, Asn468, Arg481 of the PIWI domain, and Ser288, Arg201, Lys328, His340, Lys366 of the APAZ domain. Notable hydrogen bonds to bases are from Arg243 of the MID domain, and Lys211, Arg263, Arg362 of the APAZ domain (Fig. 3). Overall, the APAZ domain fulfills a function analogous to the N, L1 and L2 regions of long pAgos in facilitating the binding of the guide

and target nucleic acid strands[5]. The 3'-end of the gRNA lies close to the APAZ-N and can putatively interact with it. A distinction from the N domain of long pAgos such as TtAgo is that APAZ-N does not block the 3'-end of the guide RNA[23]. Thus, in principle, base-pairing between the gRNA and tDNA can continue beyond the segments visible in the structure. This is consistent with findings that binding of the guide 3'-end is not required for SPARTA activation and that guide RNAs with length of even up to 50 nucleotides can activate SPARTA[3].

## Conformational changes upon nucleic acid binding

Superposition of Apo SPARTA heterodimer onto a heterodimer in the active oligomer reveals two key segments of SPARTA that change conformation to facilitate gRNA/tDNA binding. First is an extended loop between α10 and β9 (loop10-9, residues 319 to 331) of the PIWI domain that protrudes into the nucleic acid binding channel and in the cryo-EM structure reconfigures and shortens by 4 residues (to residues 323-331) to permit gRNA/tDNA duplex binding (Fig. 4a, b). Strikingly, this conformational change in loop10-9 propagates to the adjoining protein segments, wherein strand β9 loses secondary structure and attains a much more loop-like configuration and residues Asp306 and Asp309 between β8 and β9 (loop8-9) reconfigure to expose a positively charged pocket on the PIWI domain (Fig. 4a–c). Residue Asp309, for example, moves by as much as -16 Å from its position in the Apo structure. This "demasking" of a positively charged pocket on the PIWI domain (Fig. 4c) appears to be an important feature in permitting the oligomerization and activation of SPARTA on gRNA/tDNA binding. Another major conformational change is the C-terminal terminal helix (αN) of APAZ-L2 (Fig. 1f and Fig. 4a), which is ousted from its Apo position to allow binding of the 3'-end of the target DNA strand. There is no discernable density for this helix in the cryo-EM structure, suggesting its movement (and subsequent disorder) away from target DNA, and it corresponds to the region described as TIR-APAZ C-terminus in recent SPARTA papers[24–28]. Interestingly, the C2' base of the target DNA evicts from the gRNA/tDNA helix and occupies the position vacated by helix αN, leaving the second base of the guide RNA (G2) to pair with a surrogate Arg243 from the MID domain (Fig. 3a). Overall, loop10-9 and helix αN appear to be key elements that maintain SPARTA in an auto-inhibited state prior to gRNA/tDNA binding. Another striking conformational difference is in insert57 of the MID domain. Unlike the crystal structure, there is no discernable density for insert57 in the cryo-EM structure, indicative of its flexibility. In the crystal structure, insert57 is ordered and extends from the core of the MID domain (Figs. 1d and 4a) and we postulate it as putative site for interaction with other (as yet to be discovered) components of the SPARTA defense system.

## Active site

Superposition of Apo SPARTA heterodimer onto heterodimers A, B, C and D in the active oligomer reveals an asymmetric change in the TIR domains. Thus, whereas the TIR domains of heterodimers A and C are in the same orientation as in the Apo heterodimer, the TIR domains of heterodimers B and D rotate from their positions in the Apo heterodimer by -175° in the active oligomer (Supplementary Fig. 8). A hallmark of TIR domains, whether in scaffolding or enzymatic function, is self-association into oligomers[21]. For TIRs that degrade NAD+, self-association is mandatory in assembling a composite active site (spanning two subunits) for the binding and hydrolysis of NAD+. Interactions between $TIR_A$ and $TIR_C$ as well as $TIR_B$ and $TIR_D$ are mediated by a BCD interface, involving residues from helices αB and αC of $TIR_C$ or $TIR_D$ and residues from helix αD and loop DE of $TIR_A$ and $TIR_B$ (Fig. 2c). Interactions between $TIR_A$ and $TIR_B$ as well as $TIR_C$ and $TIR_D$ are by mediated by a BE interface, involving residues from the BB loop of $TIR_B$ or $TIR_D$ and residues from βD, βE and αE of $TIR_A$ or $TIR_C$ (Figs. 2c and 5a). Importantly, the residues Arg114 and Asn116 from the βE of $TIR_A$ or $TIR_C$ are involved in extensive hydrogen bonds/electrostatic interactions with residues from the BB loop of the $TIR_B$ or $TIR_D$ (Fig. 5a). To test the importance of this interface we mutated Arg114 to a glutamate and Asn116 to an alanine, evaluated the activity of SPARTA in a fluorescence assay and found that it resulted in the complete loss of activity (Fig. 5b). The same BCD and BE interfaces are observed in the similarly organized TIRs of AbTir with the active site spanning the BE interface across two subunits[22] (Fig. 2c). The BE interface is prevalent among enzymatic

TIRs, including those that are organized differently such as in SARM1[15]. Figure 5c and Supplementary Fig. 9a compare the SPARTA active site formed across $TIR_A$ and $TIR_B$ (and the same for $TIR_C$ and $TIR_D$) with that in AbTir and SARM1, the structures for which have been determined in the presence of the NAD+ analogue 3AD (8-amino-isoquinoline adenine dinucleotide)[15,22]. Many of the key residues involved in 3AD binding in AbTir and SARM1 are structurally conserved in SPARTA, including catalytic Glu77 for NAD(P)+ hydrolysis (Fig. 5c and Supplementary Fig. 9a). Additionally, we modeled NAD+ in the TIR active site (formed across $TIR_A$ and $TIR_B$) and analyzed the sequences of enzymatically active prokaryotic and eukaryotic TIR domains with CrtSPARTA$^{TIR}$ (Supplementary Fig. 9b and c). From the structural analysis, most of the active site residues are conserved among all TIRs with only a few substituted, the most intriguing of which is the substitution of Trp227 in AbTir and Trp662 in SARM1 with Tyr105 in SPARTA (Supplementary Fig. 9a). The tryptophan in AbTir and SARM1 mediates stacking with the adenine base of 3AD, and Tyr105 in SPARTA can putatively mediate the same type of aromatic stacking interactions with adenine base of NAD(P)+. Indeed, we observe significant loss in the NADase activity, when we replace Tyr105 by alanine (Fig. 5b). In AbTir, Trp204 has been shown to facilitate the generation of cyclic products from NAD+[22]. The corresponding residue in SPARTA is Trp46 and when we mutate this residue to alanine it results in almost complete loss of NADase activity (Fig. 5b), even though there is no evidence to indicate that SPARTA produces cyclic products following NAD(P)+ hydrolysis[3].

## Mechanism of oligomerization and activation

Contacts between heterodimers A and B (and similarly between C and D) in the SPARTA are dominated by symmetric protein-protein interactions between the pAgos. In particular, a positively charged pocket on $PIWI_A$ receives residues 130-135 from $MID_B$ and vice versa (Supplementary Fig. 10). Most of the interactions are polar, wherein the main chain carbonyls of residues Lys130, Asn131, Glu133, Glu134 of $MID_B$ interact with the main chain amide of Ala502 and side chains of Lys314 and Lys504 of $PIWI_A$, among other interactions. In addition, residues Gln35 and Tyr37 of $MID_B$ interact with Gly38, Lys40, Glu87 and Asn90 of $MID_A$ (Supplementary Fig. 10). Indeed, when we mutate Gln35 and Tyr37 to alanine it results in the complete loss in SPARTA enzymatic activity, indicating the importance of MID-mediated dimerization of SPARTA (Fig. 5b). Interestingly, we observe a significant population of SPARTA dimers of heterodimers in our biophysical experiments (Fig. 1c and Supplementary Fig. 2a), suggesting a hierarchy of SPARTA assembly, wherein the dimers of heterodimers form first via symmetric interactions between the pAgos and then transition to tetramers of heterodimers via interactions of their TIR domains.

A key question in understanding how SPARTA becomes activated is whether there are any conformational changes on nucleic acid binding that might promote oligomerization? Intriguingly, in Apo SPARTA, the positively charged pocket on $PIWI_A$ that mediates oligomerization is masked by negatively charged residues Asp306 and Asp309 on the loop8-9 (Fig. 4c). In the SPARTA oligomer, Asp306 and Asp309 rearrange (on loop8-9) as a consequence of the reconfiguration and shortening of loop10-9 on nucleic acid binding (vide supra, Fig. 4). Loop8-9 both lengthens and extends (towards α10) and residues Asp306 and Asp309 move >10 Å from their position in the Apo structure to demask the positively charged pocket for dimerization. We argued that deletion of residues from loop10-9 may prevent the lengthening and extension of loop8-9 and the reconfiguration of residues Asp306 and Asp309 on nucleic acid binding. Indeed, when we delete four residues from loop10-9, SPARTA loses the ability to form dimers on nucleic acid binding (Supplementary Fig. 11), lending to almost complete loss of NADase activity (Fig. 5b). Overall, reconfiguration of loop10-9 on nucleic acid binding appears to be the key

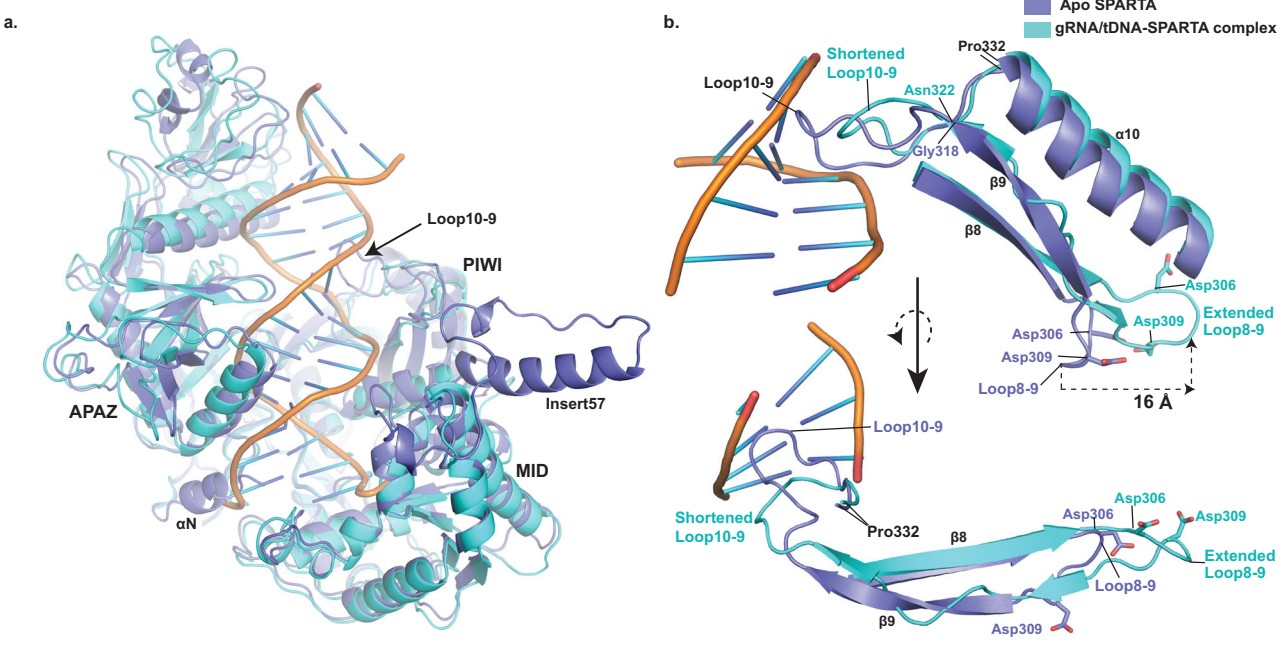

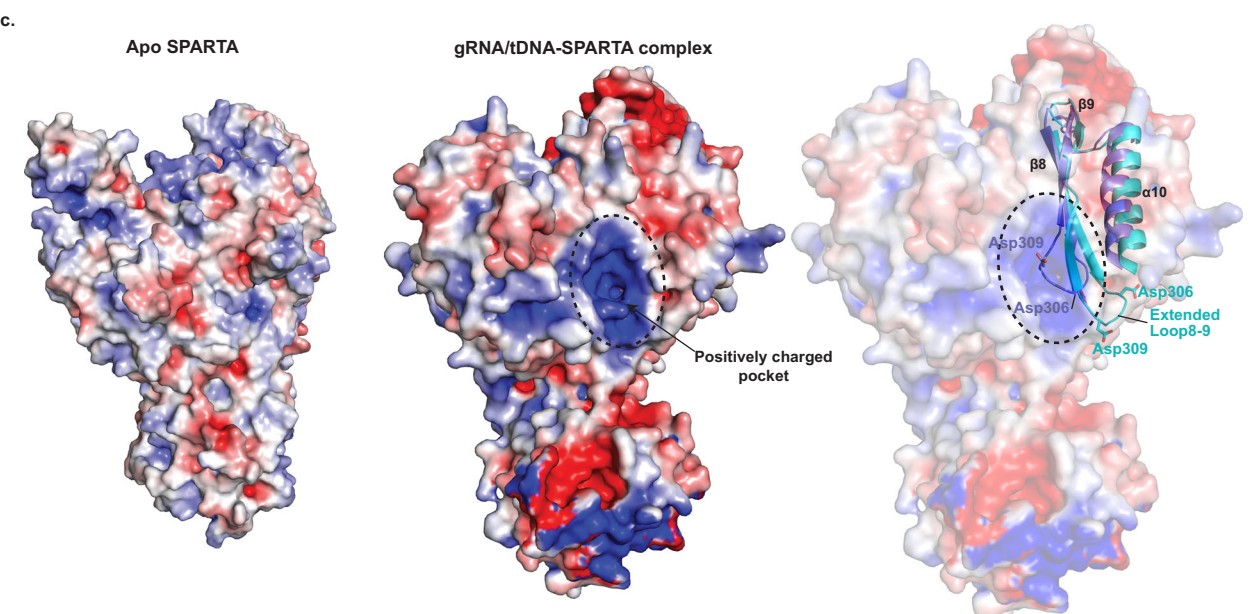

**Fig. 4 | Conformational changes upon gRNA/tDNA binding. a** Overlay of the APAZ, MID and PIWI domains in the inactive Apo and the activated SPARTA oligomer structures, wherein loop10-9 undergoes a reconfiguration and helix αN moves away from the nucleic acid binding groove to facilitate gRNA/tDNA binding. In addition, insert57 is disordered in the oligomer structure. **b** An overlay of the secondary structure elements (α10, β9 and β8), loops 10-9 and 8-9, and residues Asp306 and Asp309 between the inactive Apo and the activated SPARTA oligomer structures. Loop10-9 (between α10 and β9) undergoes reconfiguration and shortening on gRNA/tDNA binding and propagates to lengthen and extend loop8-9 (between β8 and β9) carrying residues Asp306 and Asp309. Residue Asp309 moves by as much as -16 Å from its position in the Apo structure. α10 was removed from

the lower panel for better clarity. **c** Electrostatic surface potential map of inactive Apo SPARTA (left) and active gRNA/tDNA-SPARTA complex (middle). The comparison shows the exposure of a positively charged pocket in the activated SPARTA complex. On the right, secondary structure elements, loops 10-9 and 8-9, and residues Asp306 and Asp309 from the Apo and the activated SPARTA oligomer structures overlaid on the electrostatic surface of the activated oligomer. The figure shows that the positively charged pocket on the activated oligomer surface is masked by Asp306 and Asp309 in the Apo structure. The reconfiguration and shortening of the loop10-9 upon gRNA/tDNA binding leads to the lengthening and extension of loop8-9, causing in turn the displacement of Asp306 and Asp309 from the pocket.

conformational change that propagates to the dimerization interface, as a step towards tetramerization.

During the preparation and review of our manuscript, related papers on SPARTA were published[24–31]. Our work complements and provides a firmer understanding of the SPARTA structure and mechanism. First, we succeed in deriving a crystal structure of Apo

SPARTA heterodimer that provides a high-resolution view of the TIR-APAZ protein and the MID and PIWI domains of the short pAgo. The MID domain is observed, for example, to contain a unique insertion (insert57). Second, the crystal structure permits the building of SPARTA in the cryo-EM structure of the active oligomer without reference to an AlphaFold model. Third, a comparison between the crystal and cryo-EM

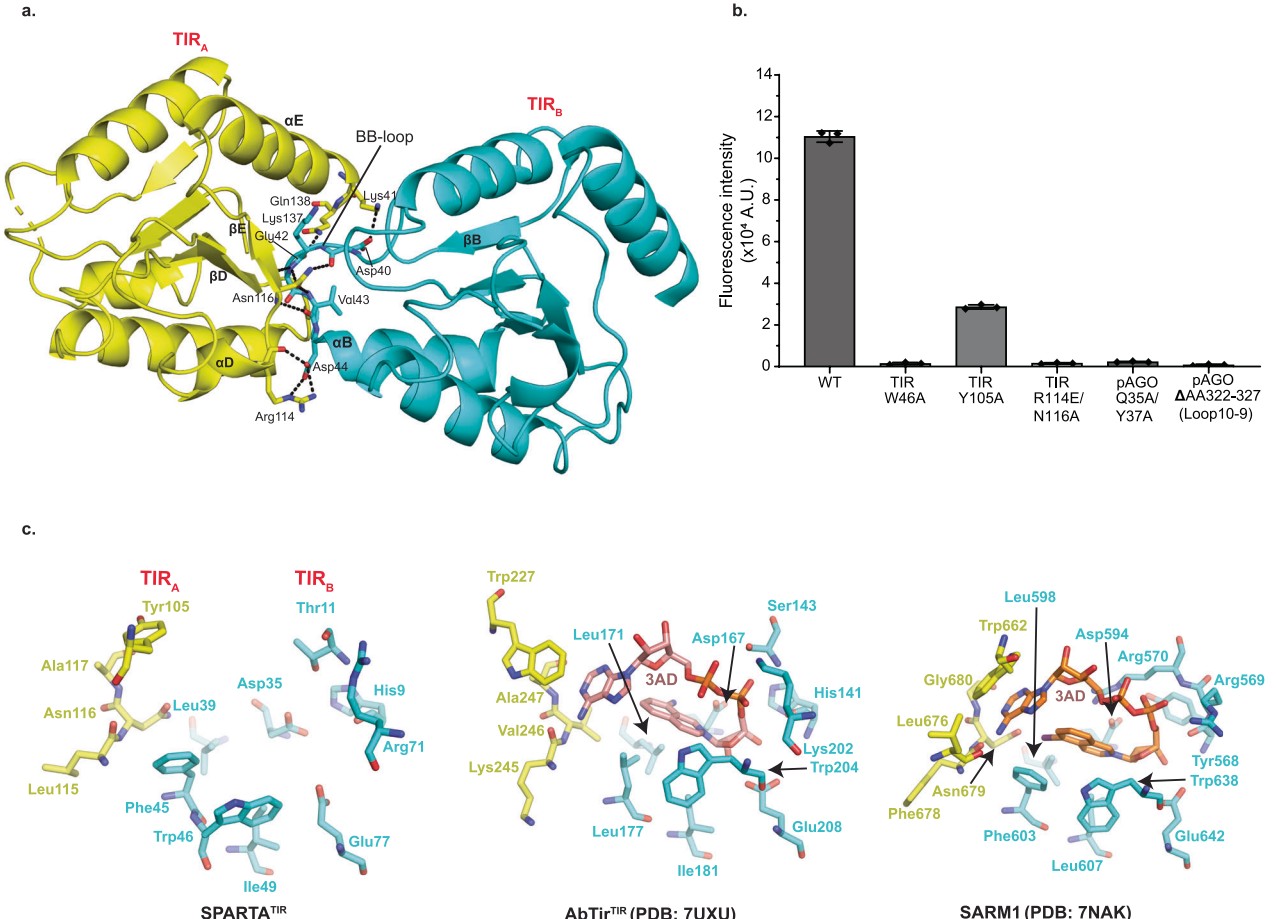

**Fig. 5 | Analysis of the TIR active site. a** Details of interactions between TIR$_A$ and TIR$_B$ (BE interface). The residues from BB-loop of TIR$_B$ forms extensive electrostatic interactions with residues from βD, βE and αE of TIR$_A$. **b** Comparison of the NADase activity of TIR and pAgo mutants with wild-type SPARTA. TIR-Trp46 and TIR-Tyr105 are catalytic residues of SPARTA which are structurally conserved in AbTir and SARM1. Arg114 and Asn116 are involved at TIR$_A$-TIR$_B$ interface. Gln35 and Tyr37 are involved at the pAgo-pAgo interface. The mutation of these residues results in a major reduction in the NADase activity. Data are presented as mean values +/- SD from experiments performed in triplicates. Source data are provided as a Source Data file. **c** Close-up view of the SPARTA active site formed across the TIR$_A$ and TIR$_B$ heterodimers is compared with that in AbTir (PDB:7UXU) and SARM1 (PDB: 7NAK). AbTir and SARM1 are depicted with bound NAD$^+$ analogue (3AD).

structures posits an activation mechanism, whereby a configurational change in a critical loop in SPARTA (loop10-9) on nucleic acid binding propagates to the oligomeric interface to create a positively charged surface for active assembly.

Collectively, the crystallographic and cryo-EM studies presented here provide a visualization of SPARTA before and after RNA/ssDNA binding, leading to NAD(P)$^+$ degradation and cell death. We show that a loop (loop10-9) on the PIWI domain and a helix (αN) on the APAZ domain reconfigure to relieve autoinhibition and permit nucleic acid binding. The transition from an inactive heterodimer to an active tetramer of heterodimer amasses the TIR domains in a parallel-strands arrangement for NAD(P)$^+$ hydrolysis. We show that reconfiguration and shortening of loop10-9 on nucleic acid binding propagates to the heterodimer dimerization interface to create a positively charged surface for active assembly. A prominent feature of the MID domain in the crystal structure is insert57, which emerges as a relatively conserved sequence feature of short pAgo systems but is absent in long pAgos (Supplementary Fig. 7). Short pAgos systems have been cataloged into four phylogenetic clades, S1B, S2A and S2B that are associated with an APAZ domain containing protein, and S1A in which SIR2/APAZ is fused to short pAgo[3,6]. SPARTA belongs to the S2A clade and proteins in the S2B and S1B clades also contain a large portion of insert57. S1A proteins contain a significant portion of insert57. We do not yet know the role of insert57. However, given its

long outward extension from the MID domain we speculate a possible role for it in interactions with host or other components of the short pAgo systems.

## Methods

### Protein purification

For the expression and purification of heterodimeric TIR-APAZ/short pAgo complex from *Crenotalea thermophila* (SPARTA), pET-6xHis-MBP-TEV-CrtTIR-APAZ/CrtAgo vector was used[3]. This vector was a kind gift from Prof. Dr. Daan Swarts (Wageningen University) (Addgene # 183146). The plasmid was transformed into *Escherichia coli* BL21 Star (DE3) cells and grown at 37 °C until the culture reached an OD$_{600}$ of ~0.8. The temperature was then reduced to 18 °C and expression induced by the addition of 0.5 mM IPTG, followed by incubation for 16 h. The cells were then resuspended in a lysis buffer (50 mM Tris pH 7.5, 250 mM NaCl, 10% glycerol, 0.01% IGEPAL, 20 mM imidazole, and 10 mM 2-mercaptoethanol) in the presence of Pierce Protease Inhibitor tablets, EDTA-free (Thermofisher), and 1 mM PMSF. The cells were lysed by sonication, clarified by centrifugation, and the filtered supernatant loaded onto a HisTrap HP affinity column (GE Healthcare), and the protein eluted using an imidazole gradient ranging from 25-500 mM. The fractions containing CrtSPARTA protein complex were pooled and further loaded onto an amylose resin column (New England Biolabs). The column was washed with 20 column volumes

(CV) of wash buffer (20 mM Tris pH 7.5, 500 mM NaCl and 5 mM 2-mercaptoethanol). The bound protein was eluted with wash buffer containing 20 mM maltose and fractions containing the protein were pooled and TEV protease was added in the 1:20 ratio for cleaving the 6xHis-MBP tag. The 6xHis-MBP tag was cleaved following dialysis against cleavage buffer (20 mM HEPES pH 7.5, 250 mM KCl, 2 mM EDTA, 5 mM 2-mercaptoethanol) at room temperature. To remove the cleaved 6xHis-MBP tag, the protein was reapplied to the HisTrap column, and the cleaved protein was collected as flow through. The flow through containing the untagged SPARTA complex was diluted to a final salt concentration of 100 mM. The diluted sample was then loaded onto a HiTrap Heparin column and the protein eluted using a KCl salt gradient from 100 mM–2.5 M. The peak fractions containing the protein of interest were pooled, concentrated, and subjected to size exclusion chromatography (SEC) using a HiLoad 16/600 Superdex 200 (GE Healthcare) column, preequilibrated with 20 mM HEPES pH 7.5, 100 mM KCl, and 1 mM DTT.

The CrtSPARTA mutants were generated by GenScript and the mutant proteins were purified in a similar fashion to the wild type protein.

### Crystallization, data collection, and structure determination

Crystallization trials for the apo SPARTA complex were carried out at a concentration of 5 mg/ml in a sitting drop vapor diffusion format, using various commercially available screen kits with 0.3 µl of protein mixed with an equal volume of reservoir solution. Small crystalline precipitate was obtained in a condition containing 0.1 M sodium acetate pH 4.5, 0.8 M sodium phosphate monobasic and 1.2 M potassium phosphate dibasic. This crystalline precipitate was used as seed for further optimization of crystals by varying the salts and seed concentration. The optimized crystals diffracted poorly (~7 Å) with synchrotron radiation under cryogenic conditions (NSLS-II 17-ID-1 and 17-ID-2 beamlines) at the Brookhaven National Laboratory. To improve diffraction, we screened various cryoprotective mixtures and observed gradual improvement in diffraction. The use of a cryoprotectant mixture containing 0.65 M L-Proline, 6.5% Glycerol, 12.5% Tacsimate pH 7.0, and 9% Sucrose resulted in an improvement in resolution to ~3 Å and the best crystal diffracted to 2.66 Å. The diffraction data were processed using autoPROC and STARANISO (Global Phasing Ltd.). The crystal belonged to P6$_5$22 space group with unit cell dimensions of a = b = 197.18 Å, c = 183.4 Å, α = β = 90°, γ = 120°. The structure was solved by molecular replacement using Phaser-MR[32] with a model of CrtSPARTA generated by AlphaFold[13] as the search model. Subsequently, iterative manual building and refinement were performed using Coot v0.8.9.1 EL[33] and Phenix v1.20cr3-4406-000 Refine[34]. All molecular graphic figures were prepared using PyMOL v2.5.4 (Schrödinger LLC).

### Reconstitution of the CrtSPARTA-gRNA/tDNA complex

The samples for mass photometry, negative staining, and single particle cryo-EM were prepared similarly. The CrtSPARTA-gRNA/tDNA duplex complex was prepared by mixing 1.5 mg/ml CrtSPARTA, 5 mM MgCl$_2$, and 16 µM 5′-phosphorylated (5′-P) 21nt-long ssRNA (5′-P-UGAGGUAGUAGGUUGUAUAGU-3′, Dharmacon) in the size exclusion buffer (20 mM HEPES pH 7.5, 100 mM KCl and 1 mM DTT). The mixture was incubated at 37 °C for 1 hour followed by addition of 16 µM 45nt-long complementary target ssDNA (5′-AAACGACGGCCAGTGCCAAGC TTACTATACAACCTACTACCTCAT-3′, IDT). SPARTA with gRNA/tDNA was then incubated at 55 °C for 12 hours. After incubation, the sample was centrifuged at 12,000 rpm for 30 min to remove any insoluble aggregates.

### Mass photometry

Mass photometry experiments were performed on a One$^{MP}$ mass photometer (Refeyn Ltd, Oxford, UK). The procedure used was based on a previously published protocol[35]. The datasets were recorded with Acquire$^{MP}$ using standard settings. Briefly, self-adhesive silicon gasket (4-6 wells, ~15 µL volume each) was placed on a clean borosilicate glass microscope cover slide. The glass slide/silicone gasket assembly was placed on the instrument's objective and centered on a single well. 12 µL of sample buffer (20 mM HEPES pH 7.5, 100 mM KCl, 2 mM DTT) was added to the well, and the focal position of the glass surface was determined and held constant using an autofocus system. 1 µL of sample (100-fold diluted Apo CrtSPARTA or the CrtSPARTA-gRNA/tDNA complex) was then added to the same well, mixed with the buffer and a 90 sec movie recorded immediately after the dilution. Contrast-to-mass linear calibration curve was created using BSA (monomer and dimer) and apoferritin protein standards prepared in the sample buffer. The movies were analyzed and processed using Discover$^{MP}$.

### Negative stain microscopy

Negatively stained samples were prepared at a sample concentration of 10 µg/ml using 400-mesh Cu/Pd grids (Ted Pella; prepared by floating carbon on it). The negatively stained samples were stained with uranyl formate and imaged on a Tecnai F20 TEM operating at 200 kV using a DE-20 camera at a calibrated magnification of 62,000X and a −2.0 µm defocus. Particle images were picked using the Blob picker module within cryoSPARC v3.3.1[36] and subjected to iterative rounds of 2-D classification.

### Cryo-EM specimen preparation and data collection

The complex of SPARTA oligomer with gRNA/tDNA was prepared on 300-mesh UltrAuFoil R1.2/1.3 µm grids. The grids were glow discharged using Ar and O$_2$ for 8 secs using solarus II plasma cleaner (Gatan) prior to loading 2.5 µL of the protein complex. Samples were prepared in a chamber with 95 % humidity at 20 °C. The grids were backblotted and vitrified in liquid ethane cooled with liquid nitrogen on the Leica GP2 plunge freezer (Leica microsystems). Vitrified samples of the SPARTA complex were screened using Glacios (Thermo-Fisher) operated at 200 kV prior to large-scale data collection. Screened grids were imaged using a Titan Krios operated at 300 kV (ThermoFisher) equipped with a BioQuantum energy filter (Gatan) and K3 direct electron detector (Gatan). The untilted (0°) and the tilted (tilted to 30° and 45°) datasets of the SPARTA complex were imaged in counting mode at a calibrated pixel size of 1.083 Å/pixel using image shift with a nominal defocus range of −0.8 µm to −2.5 µm. The frames were aligned using MotionCor2[37] in Appion[38] and CTF estimated using Patch CTF within cryoSPARC v3.3.1[36].

### Image processing of the SPARTA oligomer

Upon frame alignment and CTF estimation, particles were picked using template picker within cryoSPARC v3.3.1 using templates generated by negative stain microscopy. The initial particle stack was subjected to multiple rounds of 2-D classification which showed high resolution structural features of the complex. Selected particles were subjected to train a model (TM1) with topaz, a neural network based particle picker[39] implemented within cryoSPARC v3.3.1. The micrographs were binned by sixteen and used with resnet8 neural network architecture. Further processing of the 0° dataset resulted in anisotropic maps and, therefore, a tilted dataset (tilted to 30° and 45°, 13,884 micrographs) of the SPARTA complex was imaged. The data was collected in counting mode at 81,000X Magnification (1.083 Å/pixel). Total dose was set to 51.11 e⁻/Å$^2$ fractionated into 40 frames. Particles were reextracted from the smaller tilted dataset (6,621 micrographs) using the topaz model (TM1). Multiple rounds of 2-D classification resulted in 2-D classes which were used to train a topaz model (TM2). This model was used to pick particles from the entire tilted dataset (13,884 micrographs) and after several rounds of 2-D classification and Ab-initio clean up a new topaz model was generated (TM3). This model (TM3) was used to pick

particles from the tilted dataset (13,884 micrographs) which resulted in representative 2-D classes (Supplementary Fig. 3). The 3-D reconstruction from the selected particles resulted in an Ab-initio model (544,322 particles) which underwent 3-D classification followed by non-uniform refinement in cryoSPARC v3.3.1[40] and resulted in an isotropic construction with a sphericity of 0.90. The reconstruction was generated using 238,432 particles and had a nominal resolution ($FSC_{0.143}$) of 3.35 Å (Supplementary Fig. 3).

Each monomer in the 'wings' (APAZ-MID-PIWI-RNA/DNA) is related to the other by C2 symmetry. In order to improve the resolution of the 'wings' of the SPARTA oligomer complex, focused refinement of each of the wings (dimers) was performed separately. We fitted the modeled coordinates into the globally refined map (described below) and generated a mask using the command molmap in UCSF chimera (Supplementary Fig. 4). In order to perform the C2-focused refinement of the 'wings', the approximate voxel coordinates of the symmetry center were found using the center of mass of the mask for each of the 'wings' using UCSF chimera. This center was used to translate the volume, mask, and stack of particles with C2 symmetry enforced (volume alignment utility) followed by local refinement in cryoSPARC v3.3.1 (Supplementary Fig. 4). C2-focused refinement improved the nominal resolution of each of the 'wings' to 3.15 Å and 3.17 Å, respectively and resulted in higher local resolution maps for them (Supplementary Figs. 4 and 5). Local refinement of the TIR domains was performed with C1 symmetry.

### Model building, refinement, and analysis of the SPARTA oligomer

A model of the apo form of the SPARTA heterodimer from the crystal structure was used to build into the initial globally refined cryo-EM density map of the complex of SPARTA oligomer (with guide RNA and target DNA) by rigid body docking in Coot v0.9.3. Coot was used to build the gRNA/tDNA and manually rebuild and reorganize sections of the tetramer which were different from that of the monomer. There is some putative density for A1' of target DNA but due to its weak nature, we did not build this nucleotide. A composite map obtained from focused refinements was generated using the vop maximum command in UCSF Chimera. The model was rebuilt and refined against the composite map using multiple rounds of real space refinement routine within Phenix v1.20[41] with both base-pair and secondary structure restraints imposed. The models were validated using MolProbity[42]. Superimposition of structures was performed in Coot. The figures were prepared using UCSF ChimeraX v1.3[43] and PyMOL v2.5.4.

### ε-NAD assay

A reaction mixture consisting of purified SPARTA heterodimer or the SPARTA mutants in SEC buffer, 50 μM ε-NAD$^+$ (Biolog, N010), 2 μM 21nt-long ssRNA guide and 5X reaction buffer (50 mM MES pH 6.5, 375 mM KCl, and 10 mM MgCl2) was prepared in PCR tubes. The mixture was incubated at 37°C for 15 min, after which a 45nt-long ssDNA target was added to a final concentration of 2.5 μM. The final concentrations of each component were – 1 μM SPARTA complex or the mutants, 50 μM ε-NAD$^+$, 10 mM MES pH 6.5, 125 mM KCl, and 2 mM MgCl$_2$ in a final volume of 60 μL. After the addition of the target DNA, the tubes were transferred to a preheated PCR machine at a temperature of 55°C and incubated for 1 hour. The fluorescence intensity was measured using an excitation wavelength of 310 nm and an emission wavelength of 410 nm using a Synergy H1 Hybrid Multi-Mode Plate Reader. The graph was generated using GraphPad Prism v10.0.2 (232).

### Reporting summary

Further information on research design is available in the Nature Portfolio Reporting Summary linked to this article.

## Data availability

The structure factors and coordinate files for the crystal structure of the Apo SPARTA have been deposited in the Protein Data Bank (PDB) under the accession code 8U7B.

The original, composite cryo-EM density map as well as the focused refinement maps of the SPARTA oligomer generated in this study have been deposited in the Electron Microscopy Data Bank (EMDB) under accession numbers EMD-41959, (Original map), EMD-41945 (Focused refined map of the dimer [APAZ-MID-PIWI-RNA/DNA]), EMD-41947 (Focused refined map of the second dimer [APAZ-MID-PIWI-RNA/DNA]), EMD-41948 (Focused refined map of the TIR domains) and EMD-41966 (Composite map). The resulting atomic coordinates for the SPARTA oligomer have been deposited in the Protein Data Bank (PDB) with accession number PDB ID: 8U72.

Raw uncropped gel image (for Fig. 1b) and raw ε-NAD$^+$ assay data (for Fig. 5b) are available as source data file accompanying this manuscript. Source data are provided in this paper.

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

## Acknowledgements

This work was funded by grant R35-GM131780 (A.K.A) from the National Institutes of Health (NIH). We thank the staff at the National Synchrotron Light Source II (NSLS-II) beamlines 17-ID-1 and 17-ID-2 for facilitating X-ray data collection. NSLS-II is a United States Department of Energy (DOE) Office of Science User Facility operated for the DOE Office of Science by Brookhaven National Laboratory under Contract No. DE-SC0012704. The Center for BioMolecular Structure (CBMS) at NSLS-II is primarily supported by the NIH, National Institute of General Medical Sciences (NIGMS) through a Center Core P30 Grant (P30GM133893), and by the DOE Office of Biological and Environmental Research (KP1605010). Mass photometry experiments and most of the cryo-EM work were performed at the Simons Electron Microscopy Center and National Resource for Automated Molecular Microscopy, located at the New York Structural Biology Center, supported by grants from the Simons Foundation (SF349247), NYSTAR, and the NIH National Institute of General Medical Sciences (GM103310), with additional support from Agouron Institute (F00316), NIH (OD019994) and NIH (RR029300). We thank Dr. William (Chase) Budell for the assistance with the mass photometry experiments. We also thank Carolina Hernandez and Joshua Mendez for their assistance with cryo-EM data collection and Oliver Clarke for his input on cryo-EM data processing.

Computing resources needed for this work were provided in part by the High-Performance Computing facility of the Icahn School of Medicine at Mount Sinai. Molecular graphics and analyses were performed with UCSF Chimera, developed by the Resource for Biocomputing, Visualization, and Informatics at the University of California, San Francisco, with support from NIH P41-GM103311.

## Author contributions

J.K. and A.K.A. conceived the project; J.K., R.M. and A.K.A. designed the experiments; J.K performed the protein expression, activity assays, and X-ray crystallographic studies; J.K. and R.M. performed the mass photometry studies; R.M. performed the single particle EM studies, assisted by J.K.; A.K.A. guided the overall project. A.K.A., J.K. and R.M. wrote the manuscript.

## Competing interests

The authors declare no competing interests.
