## [Peer Review File · Nature Communications]

Nucleic Acid Mediated Activation of a Short Prokaryotic Argonaute Immune SystemREVIEWER COMMENTS

Reviewer #1 (Remarks to the Author):

Kottur, Malik, and Aggarwal, describe structures of the short prokaryotic Argonaute (pAgo) TIR-APAZ system (SPARTA). Using both cryo-EM and X-ray crystallography, the authors detail the overall architecture of the SPARTA system. In detail, the authors describe the disposition of each subunit and present models for the activation of the NADP hydrolysis activity. Overall, the authors describe the structures well, but the manuscript is limited in scope without in vitro experiments to support the models they present. Undoubtedly the manuscript makes a timely contribution to the field of research around bacterial defense systems. However, it cannot be ignored that studies have been very recently published that are in contrast, deeper in mechanistic detail. It is the opinion of this reviewer that it is to the editor's discretion whether the manuscript is within scope of the journal given the current state of the field.

Major Comments

1) To support the models presented in the manuscript, the authors would do well to carry out experiments (phage or biochemical assays) where key residues are mutated to enhance the description of the structures throughout the main text. In particular, the authors ascribe a key role to a region named insert 57 but present no evidence beyond its varied disposition within the structures. The authors also suggest that insert 57 plays a role in an interaction with other components of the SPARTA system. This feels like an unsubstantiated claim and lacks context. Could the authors clarify how they have come to this conclusion?

2) The authors describe how SPARTA becomes activated and whether conformational changes upon nucleic acid binding might promote oligomerization. This is a very interesting point that is not directly addressed in the recent work from other similar studies. In the context of this study, the authors state that the loop 10-9 within the PIWI domain and an alpha helix on the APAZ domain undergo conformational changes to relieve autoinhibition and permit nucleic acid binding. However, the recent papers suggest that autoinhibition is alternatively governed by the TIR-APAZ C-terminus. Could the authors compare their observations and this model?

General Comments

Overall, the authors would do well to simplify figures and make them more internally consistent, especially around labelling and highlighting what feel like unnecessary details. Some of the labelling throughout seems excessive and somewhat detracts from the key messages that the authors may want to get across to the reader. E.g. Fig.1 needs some minor changes to improve the clarity and visibility of labels. It is also unclear why the authors have highlighted each individual secondary structure element in the figure.

Page 3, Line 62. The SPARTA system was first characterized in 2022 but was first discovered in 2018 (<https://pubmed.ncbi.nlm.nih.gov/30563906/>)

Reviewer #2 (Remarks to the Author):

In this work, Kottur, Malik, and Aggarwal investigate the structural basis for NAD⁺ cleavage activity of the short prokaryotic argonaute TIR-APAZ (SPARTA) system. The authors resolved the structures of an inactive heterodimer of the nucleic acid binding short pAgo subunit and the NAD⁺ degrading TIR-APAZ subunit, along with the active tetramer of the previously resolved heterodimer unit in complex with an RNA guide duplexed with a target ssDNA via crystallography and cryo-EM, respectively. Interestingly, the structures provide an explanation for activity by the TIR domains in the nucleic acid duplex-bound state. Specifically, this bound state facilitates the oligomerization of the inactive heterodimer unit into an active tetramer of heterodimers, creating contacts between multiple TIR domains that poise them for NAD⁺ cleavage activity. The authors then go on to highlight the key contacts between residues in the pAgo and the APAZ domain of TIR-APAZ that mediate binding of guide RNAs and a target DNAs and connect how these contacts lead to the oligomerization of the heterodimer units. Finally, NAD⁺ cleavage by active SPARTA was demonstrated via a fluorescence from a cleaved NAD⁺ analog that fluoresces brighter when cleaved, and it was shown that this activity is dependent on putative catalytic residues within the TIR domain along with residues that seem to facilitate dimerization between pAgos. This study not only provides a convincing explanation for the activation and catalytic activity of SPARTA, it also expands on the mechanistic diversity of TIR domain oligomerization translating to catalytic activity in bacterial immunity.

Although this reviewer recognizes the value of the findings within this study and their impact on the field of bacterial immunity, the authors obtained a considerable amount of information about the activity of SPARTA that should be further investigated to validate the findings. Only panel 4b challenges the structural results through mutational analysis. Residues involved in complex formation, guide binding, conformational changes, oligomerization and activation, should be explored through mutagenesis.

Other comments:

- investigating the kinetics of NAD⁺ cleavage between WT active and catalytic inactive SPARTA mutants would provide greater insights into the oligomerization activity of SPARTA heterodimers and downstream NAD⁺ cleavage activity. In addition, I find it important to provide more detailed binding model of NAD⁺ into the active site of the catalytic TIR domain dimer unit.

-The authors provide models of the putative active site of active TIR dimers in figures 4a and c. Additionally, they show homologous TIR active site structural models in figure 4c to support their hypothesis of how active site residues contribute to NAD⁺ cleavage. I believe it is important to investigate this further, ideally with a structure bound to the 3AD NAD⁺ analog. In lieu of this, sequence alignments between catalytic TIR domains and TIR-APAZ in addition to docking models of 3AD to the catalytic TIR dimer would corroborate the catalytic activity observed and provide a more complete understanding of how the active site recognizes its substrate and mediates activity.

-For the reconstitution of the CrtSPARTA-gRNA/tDNA complex it is noted that the incubation for cryo-EM and mass photometry is over the course of 12 hours at 55 degrees. I believe it is important to provide greater context into kinetics behind this as well as the catalytic activity of the oligomeric complex. Fluorescence from NAD⁺ cleavage could be monitored prior to and after the addition of target DNA in preformed CrtSPARTA-gRNA complex at multiple temperatures and with WT complex along with the Q35A/Y37A pAgo mutant. This would provide insights into the speed of formation of the oligomer of heterodimers, the temperature dependence of this process, and corroborate the importance of Q35A and Y37A in promoting multimerization. Finally, performing the same experiment in 4b over a long time course could provide greater information on whether the mutations tested completely abolish activity or partially hinder activity over time.

- readers would benefit from seeing structural alignments of single protein domains across some of the homologous domains mentioned in the text and figures as additional figures. Specifically, aligning the SPARTA TIR domain with each of the other TIR domains shown in Figure 1e would be helpful to better appreciate the structural similarities and differences. Doing the same for 4c would also be helpful.

-In figure 4b, the double mutations listed under their respective bars in the bar graph should be separated somehow for readability. For example, R114EN116A should be changed to R114E/N116A or R114E + N116A.

Point-by-Point Response to Reviewers

Reviewer #1:

Reviewer #1 notes that "...the manuscript makes a timely contribution to the field of research around bacterial defense systems." but would like to see additional in-vitro experiments.

Below we address the reviewer's concerns/queries:

Query 1:

"However, it cannot be ignored that studies have been very recently published that are in contrast, deeper in mechanistic detail."

Response:

The reviewer is referring to related papers on SPARTA that were very recently published in *Nature* (Shen et al. (2023) 7 September) and *Sci. Adv.* (Ni et al. (2023) 19 July), among others. These are important papers but we disagree with the reviewer that they are deeper in mechanistic detail than our manuscript. To the contrary, our manuscript is the first to report a high-resolution crystal structure of Apo SPARTA heterodimer. This lends to much more detailed architectural view of the TIR-APAZ protein and the MID and PIWI domains of the short pAgo, wherein the MID domain is observed, for example, containing a unique insertion (insert57; described further below). The crystal structure also allows us to build SPARTA in the cryo-EM structure of the active oligomer directly without any reference to an AlphaFold model. This combined with tilted cryo-EM data for greater sphericity and local refinement of symmetric regions in the cryo-EM map allow us to build an accurate model of the active oligomer. At the same time, a direct comparison between the crystal and cryo-EM structures has allowed us to posit a mechanism of activation, whereby a configurational change in a critical loop in SPARTA (loop10-9) on nucleic acid binding propagates to the oligomeric interface to create a positively charged surface for active assembly. We believe this to be the key in understanding the promotion to active assembly and as the reviewer notes "This is a very interesting point that is not directly addressed in the recent work from other similar studies". We firmly believe that the novel structural features and the activation mechanism that we have uncovered complements the recent studies and provides a deeper understanding of the SPARTA structure and mechanism.

Query 2:

"To support the models presented in the manuscript, the authors would do well to carry out experiments (phage or biochemical assays) where key residues are mutated to enhance the description of the structures throughout the main text. In particular, the authors ascribe a key role to a region named insert 57 but present no evidence beyond its varied disposition within the structures. The authors also suggest that insert 57 plays a role in an interaction with other components of the SPARTA system. This feels like an unsubstantiated claim and lacks context. Could the authors clarify how they have come to this conclusion?"

Response:

We do carry out a significant number of mutational studies in the manuscript to test many different aspects of the structure and the results are presented in Figure 4b. These mutations include residues Arg114 and Gln116 observed at the TIR-TIR interface of the active oligomer, residues Tyr104 and Trp46 observed in the active site, and residues Gln35 and Tyr37 observed at the MID-MID interface. The structural role of these residues and the basis for their mutation are described on pages 11 and 12 of the manuscript. The reviewer suggests that more can be done

to enhance the manuscript. In the revised manuscript, we include substantial new in vitro data. This involves the deletion of residues (322-327) from loop10-9 which protrudes into the nucleic acid binding channel and undergoes reconfiguration and shortening upon binding to gRNA/tDNA. This process exposes the negatively charged residues for oligomerization. Deletion of residues from this loop leads to the loss of NADase activity, corroborating the significance of the reconfiguration and shortening of loop 10-9 in the activation of SPARTA. We include these new data in a revised Fig. 4b and describe it on pages 12-13, namely “And, when we delete residues from the loop10-9 (residues 322-327), it results in almost the complete loss of NADase activity (Fig. 4b), corroborating the significance of the reconfiguration and shortening of loop 10-9 in the activation of SPARTA.”

As for Insert57, it is one of the unique features of the MID domain that emerges from our crystal structure of the Apo SPARTA heterodimer (and had not previously been seen in other MID domain structures). It is a large helical subdomain (52 residues) that extends from the body of the SPARTA MID domain. The fact that it is not visible in the cryo-EM structure indicates its overall flexibility. Based on the reviewer’s comment, we looked for the presence of Insert57 in other short pAgos, as well as in long pAgos. Strikingly, from our sequence alignments, Insert57 emerges as a conserved sequence feature of short pAgos but not of long pAgos. In the revised manuscript, we include a new figure (Supplementary Fig. 5) that shows the sequence alignments. And on page 7 we have modified the text to “A unique feature of the SPARTA MID domain is “Insert57” found between $\alpha 5$ and $\alpha 7$, composed of a long helix ($\alpha 6$) and two short helices, that extends outwards from the body of the MID domain. Intriguingly, from sequence alignments, Insert57 emerges as a strongly conserved feature of short pAgos but not of long pAgos (Supplementary Fig. 5).”

We do not yet know the function of insert 57 but its occurrence in the crystal structure and relative conservation in other short pAgos (but not long pAgos) hints at its important function in short pAgo biology. The best we can do at this stage is to highlight its presence in the SPARTA MID domain and speculate on its possible role in protein-protein interactions. In the revised manuscript, on pages 13-14, we further expand to “A prominent feature of the MID domain in the crystal structure is Insert57, which emerges as a relatively conserved sequence feature of short pAgo systems but is absent in long pAgos (Supplementary Fig. 5). Short pAgos systems have been cataloged into four phylogenetic clades, S1B, S2A and S2B that are associated with an APAZ domain containing protein, and S1A in which SIR2/APAZ is fused to short pAgo^{3,6}. SPARTA belongs to the S2A clade and proteins in the S2B and S1B clades also contain a large portion of Insert57. S1A proteins contain a significant portion of Insert57. We do not yet know the role of Insert57. However, given its long outward extension from the MID domain we speculate a possible role for it in interactions with host or other components of the short pAgo systems.”

Query 3:

“The authors describe how SPARTA becomes activated and whether conformational changes upon nucleic acid binding might promote oligomerization. This is a very interesting point that is not directly addressed in the recent work from other similar studies. In the context of this study, the authors state that the loop 10-9 within the PIWI domain and an alpha helix on the APAZ domain undergo conformational changes to relieve autoinhibition and permit nucleic acid binding. However, the recent papers suggest that autoinhibition is alternatively governed by the TIR-APAZ C-terminus. Could the authors compare their observations and this model”

Response:

The reviewer is correct in noting the novelty of some of the conformational changes we observe on nucleic acid binding. In particular, the conformational change of loop 10-9 within the PIWI

domain on nucleic acid binding and the mechanism by which it then promotes oligomerization is an exciting feature of our work. We also describe a conformational change in an alpha helix (α N) on the APAZ domain to relieve autoinhibition and permit nucleic acid binding. This conformational change in the alpha helix is the same as that described in other recent papers. It is the same region, namely at the C-terminal region of the APAZ domain. We refer to this C-terminal region as an alpha helix (α N), due to its better definition in the crystal structure whereas the other recent papers refer to it as TIR-APAZ C-terminus. To make this clearer, in the revised manuscript (page 10), we have modified the text to “There is no discernable density for this helix in the cryo-EM structure, suggesting its movement (and subsequent disorder) away from target DNA, and it corresponds to the region described as TIR-APAZ C-terminus in recent SPARTA papers²⁴⁻²⁸.”

Query 4:

“Overall, the authors would do well to simplify figures and make them more internally consistent, especially around labelling and highlighting what feel like unnecessary details. Some of the labelling throughout seems excessive and somewhat detracts from the key messages that the authors may want to get across to the reader. E.g. Fig.1 needs some minor changes to improve the clarity and visibility of labels. It is also unclear why the authors have highlighted each individual secondary structure element in the figure.”

Response:

Based on the reviewer’s suggestion, we have simplified the labeling on some of the figures. For Fig.1, for example, we have changed the font size of the labels to make them consistent and clearly visible. Importantly, the determination of Apo SPARTA heterodimer crystal structure has allowed us to define these structural elements accurately and provides a framework for comparison to previous Argonaute crystal structures (PMIDs: 28319084 and 16061186).

Query 5:

“Page 3, Line 62. The SPARTA system was first characterized in 2022 but was first discovered in 2018 (<https://pubmed.ncbi.nlm.nih.gov/30563906/>)”

Response:

We thank the reviewer for noting this omission. In the revised manuscript (page 3), we include the modified sentence “The SPARTA system was first discovered in 2018⁶ and further characterized by Koopal et al³ in 2022...”

We thank the reviewer for the very helpful comments, which have greatly helped to improve the quality of the manuscript.

Reviewer #2:

The reviewer notes the novelty and importance of our study, stating that “This study not only provides a convincing explanation for the activation and catalytic activity of SPARTA, it also expands on the mechanistic diversity of TIR domain oligomerization translating to catalytic activity in bacterial immunity.”

We address below the reviewer’s comments/queries:

Query 1:

“Although this reviewer recognizes the value of the findings within this study and their impact on the field of bacterial immunity, the authors obtained a considerable amount of information about the activity of SPARTA that should be further investigated to validate the findings. Only panel 4b

challenges the structural results through mutational analysis. Residues involved in complex formation, guide binding, conformational changes, oligomerization and activation, should be explored through mutagenesis.”

Response:

The mutational analyses presented in Fig. 4b tests many key ideas about the TIR-TIR interface (residues Arg114 and Gln116), MID-MID interface (residues Gln35 and Tyr37) and the active site (residues Tyr104 and Trp46) upon oligomerization and nucleic acid binding. The reviewer suggests that more can be done. Based on the reviewer’s suggestion, we undertook a mutational experiment to test one of the most important results of our study, namely the reconfiguration of loop10-9 to promote active assembly. When we delete residues (322-327) from loop10-9, it abolishes NADase activity, corroborating the significance of the loop in the activation of SPARTA. We include these new data in a revised Fig. 4b and describe it on pages 12-13, namely “And, when we delete residues from the loop10-9 (residues 322-327), it results in almost the complete loss of NADase activity (Fig. 4b), corroborating the significance of the reconfiguration and shortening of loop 10-9 in the activation of SPARTA.”

Query 2:

“...investigating the kinetics of NAD⁺ cleavage between WT active and catalytic inactive SPARTA mutants would provide greater insights into the oligomerization activity of SPARTA heterodimers and downstream NAD⁺ cleavage activity. In addition, I find it important to provide more detailed binding model of NAD⁺ into the active site of the catalytic TIR domain dimer unit.”

Response:

Based on the reviewer’s suggestion, we include a new figure that docks NAD⁺ into the active site (Supplementary Fig. 7b)

Query 3:

“The authors provide models of the putative active site of active TIR dimers in figures 4a and c. Additionally, they show homologous TIR active site structural models in figure 4c to support their hypothesis of how active site residues contribute to NAD⁺ cleavage. I believe it is important to investigate this further, ideally with a structure bound to the 3AD NAD⁺ analog. In lieu of this, sequence alignments between catalytic TIR domains and TIR-APAZ in addition to docking models of 3AD to the catalytic TIR dimer would corroborate the catalytic activity observed and provide a more complete understanding of how the active site recognizes its substrate and mediates activity.”

Response:

To further investigate the active site it would be ideal to determine the cryo-EM structure of the SPARTA oligomer bound to the 3AD NAD⁺ analog. In lieu of this, as suggested by the reviewer, we undertook a sequence alignment of the catalytic TIR domains of prokaryotic and eukaryotic origin and docking studies with NAD⁺ to better understand how the active site recognizes its substrate and mediates activity. The docked model and the alignment are presented in new Supplementary panels Fig. 7b and 7c, respectively. We have added a discussion about these studies in the page 11 as follows: “Additionally, we modeled NAD⁺ in the TIR active site (formed across TIR_A and TIR_B) and analyzed the sequences of enzymatically active prokaryotic and eukaryotic TIR domains with CrtSPARTA^{TIR} (Supplementary Fig. 7b and 7c). From the analysis most of the active site residues are conserved among all TIRs with only a few substituted, the most intriguing of which is the substitution of Trp227 in AbTir and Trp662 in SARM1 with Tyr105 in SPARTA.”

Query 4:

“For the reconstitution of the CrtSPARTA-gRNA/tDNA complex it is noted that the incubation for cryo-EM and mass photometry is over the course of 12 hours at 55 degrees. I believe it is important to provide greater context into kinetics behind this as well as the catalytic activity of the oligomeric complex. Fluorescence from NAD⁺ cleavage could be monitored prior to and after the addition of target DNA in preformed CrtSPARTA-gRNA complex at multiple temperatures and with WT complex along with the Q35A/Y37A pAgo mutant. This would provide insights into the speed of formation of the oligomer of heterodimers, the temperature dependence of this process, and corroborate the importance of Q35A and Y37A in promoting multimerization. Finally, performing the same experiment in 4b over a long time course could provide greater information on whether the mutations tested completely abolish activity or partially hinder activity over time.”

Response: We agree that it would be interesting to explore the kinetics but a major complication is that the fluorescent ϵ -NAD⁺ analog used by us and all other laboratories in the field is prone to degradation even at an ambient temperature (Biolog manufacturer’s catalog # N010) and more so at higher temperatures. Indeed, we see this instability when we monitor the fluorescence from the ϵ -NAD⁺ analog over a 12h period at 55°C. There is a linear increase in fluorescence over time indicating breakdown ϵ -NAD⁺ into NAM and fluorescent ϵ -ADPR. This instability of ϵ -NAD⁺ makes it challenging to interpret temperature and time dependent experiments (at least at present). The thrust of our manuscript is to convey the structure and mechanism as an inspiration for complementary experiments in laboratories throughout the world.

Query 5:

“...readers would benefit from seeing structural alignments of single protein domains across some of the homologous domains mentioned in the text and figures as additional figures. Specifically, aligning the SPARTA TIR domain with each of the other TIR domains shown in Figure 1e would be helpful to better appreciate the structural similarities and differences. Doing the same for 4c would also be helpful.”

Response:

Based on the reviewer’s suggestion, we have now added the structural alignments of TIR, pAgo and APAZ domains as an additional figure in Supplementary Fig. 4a, 4b and 4c respectively. The structural alignment of TIR active site was included as Supplementary Fig. 7a.

Query 6:

“In figure 4b, the double mutations listed under their respective bars in the bar graph should be separated somehow for readability. For example, R114EN116A should be changed to R114E/N116A or R114E + N116A.”

Response:

We thank the reviewer for this helpful suggestion. In the revised manuscript, we have modified the labels accordingly.

We thank the reviewer for the very helpful comments, which have greatly helped to improve the quality of the manuscript.

REVIEWER COMMENTS

Reviewer #1 (Remarks to the Author):

In revision, Kottur, Malik, and Aggarwal have updated the manuscript and addressed some of the reviewers concerns.

One outstanding question that remain unresolved is understanding how SPARTA becomes activated. This is a key point of the authors manuscript as reflected in the title, abstract, and conclusions. However, to my interpretation, the structural and biochemical data presented do not reflect the same conclusion. The authors describe loop 10-9 in Apo-SPARTA as masking the oligomerization site of PIWIa. Upon nucleic acid binding, a re-arrangement exposes this site (loop reconfiguration and shortening) to expose the dimerization interface. To validate this model, the authors deleted loop 10-9 and observed a complete loss of NADase activity of the TIR domain. The authors state that these data are "corroborating the significance of the reconfiguration and shortening of loop 10-9 in the activation of SPARTA." How does the loss of NADase activity support the authors model that conformational change in this region is important for activation? Did the authors assess if the deleted 10-9 mutant was able to form dimers? What exactly is the nature of the defect in the NADase activity? I would expect (provided that the deletion has not structurally compromised the complex) that the exposure of the positively charged cleft would drive dimerization in the absence of nucleic acid and potentially even activation of the system. Or is this just a case of a non-functional protein because of misfolding? With the data presented it is impossible to tell.

As per the previous revision, it is the opinion of this reviewer that it is to the editor's discretion whether the manuscript is within scope of the journal given the current state of the field.

Reviewer #2 (Remarks to the Author):

The authors have added an experiment where different mutations were made to further test their model, which have corroborated and expanded the previous work. In the opinion of this reviewer the manuscript is ready for publication

Point-by-Point Response to Reviewers

Reviewer #1:

Reviewer #1 has one outstanding question/query, which we address below:

Query:

“One outstanding question that remain unresolved is understanding how SPARTA becomes activated. This is a key point of the authors manuscript as reflected in the title, abstract, and conclusions. However, to my interpretation, the structural and biochemical data presented do not reflect the same conclusion. The authors describe loop 10-9 in Apo-SPARTA as masking the oligomerization site of PIWIa. Upon nucleic acid binding, a re-arrangement exposes this site (loop reconfiguration and shortening) to expose the dimerization interface. To validate this model, the authors deleted loop 10-9 and observed a complete loss of NADase activity of the TIR domain. The authors state that these data are "corroborating the significance of the reconfiguration and shortening of loop 10-9 in the activation of SPARTA." How does the loss of NADase activity support the authors model that conformational change in this region is important for activation? Did the authors assess if the deleted 10-9 mutant was able to form dimers? What exactly is the nature of the defect in the NADase activity? I would expect (provided that the deletion has not structurally compromised the complex) that the exposure of the positively charged cleft would drive dimerization in the absence of nucleic acid and potentially even activation of the system. Or is this just a case of a non-functional protein because of misfolding? With the data presented it is impossible to tell”

Response:

We thank the reviewer for the query and agree fully that understanding how SPARTA becomes activated is a key question in the study of SPARTA. To our knowledge, none of the other published studies have actually addressed this pertinent question – and the reason we are able to approach it is because of the high-resolution crystal structure of Apo SPARTA heterodimer that provides the basis for a robust comparison to the cryo-EM structure of the SPARTA oligomer. Remarkably, this comparison reveals the unmasking of a positively charged pocket on the PIWI surface right at the site of dimerization. In response to the reviewer’s query, we have made numerous changes to the manuscript and include new mass photometry data, as detailed below:

1) We include a new Figure (Figure 4) that better highlights and elaborates on the conformation changes upon gRNA/tDNA binding which leads to the unmasking of the positively charged pocket. Some of these details were illustrated previously in supplementary figures and hard to find. But given the sheer importance of this point, as noted by the reviewer, we have redrawn and present a new main Figure 4, which shows the unmasking of the positively charged pocket and the accompanying conformational changes much more clearly.

2) Our cursory description of the conformation changes in the main text was clearly inadequate and led to some confusion. For example, it is not loop10-9 in Apo-SPARTA that is masking the positively charged pocket (or the oligomerization site) but residues Asp306 and Asp309 on an adjacent segment (loop8-9). Upon gRNA/tDNA binding, the conformational change in loop10-9 propagates to the adjoining segments, such that loop8-9 carrying Asp306 and Asp309 is majorly shifted and Asp309, for example, moves by as much as ~16 Å from its position in the Apo structure to expose the positively charged pocket for oligomerization. In the revised manuscript, we have expanded the text to explain this in much better detail, namely (on page 10) “Strikingly, this conformational change in loop10-9 propagates to the adjoining protein segments, wherein strand β 9 loses secondary structure and attains a much more loop-like configuration and residues Asp306 and Asp309 between β 8 and β 9 (loop8-9) reconfigure to expose a positively charged pocket on the PIWI domain (Figs. 4a-c). Residue Asp309, for example, moves by as much as

~16Å from its position in the Apo structure. This “demasking” of a positively charged pocket on the PIWI domain (Fig. 4c) appears to be an important feature in permitting the oligomerization and activation of SPARTA on gRNA/tDNA binding.”

3) Interestingly, on gRNA/tDNA binding, loop10-9 reorients and shortens whilst loop8-9 extends and lengthens. These changes appear to be highly coupled in that slack from the shortening of loop10-9 is taken up by loop8-9 for its lengthening and extension. Based on this coupling, we considered the idea of removing some slack from loop10-9 by deleting some residues from it that may preclude the lengthening and extension of loop8-9 and the subsequent reconfiguration of residues Asp306 and Asp309 on nucleic acid binding. The expectation was that it would cause a loss in the ability of SPARTA to form dimers on nucleic acid binding and the consequent loss of NADase activity. In the previous version of the manuscript, we reported the loss of NADase activity for the deletion mutant but had not shown a loss in the ability to form dimers (as enquired by the reviewer). Based on the reviewer’s comment, we conducted a mass photometry experiment on the deletion mutant and confirm a major loss in the ability of the mutant to form dimers upon gRNA/tDNA binding. There is no indication that the mutant is misfolded as it purifies in exactly the same manner as the wild type (wt) protein and it elutes from the gel filtration column at exactly the same position as the wt protein and there is no peak indicative of any aggregation (as would be expected for a misfolded protein). Additionally, the mutant protein retains the ability to bind to the gRNA/tDNA, as demonstrated by the mass photometry data (MW of ~134.6 kDa, close to the expected MW of 132.6 kDa) suggesting that the protein is folded. In the revised manuscript, we include the new mass photometry data on the deletion mutant in a new supplementary Figure 9. Also, we have revised the text to indicate the new data and provide a better basis for making the deletion mutant, namely (on page 13) “Intriguingly, in Apo SPARTA, the positively charged pocket on PIWI_A that mediates oligomerization is masked by negatively charged residues Asp306 and Asp309 on the loop8-9 (Fig. 4c). In the SPARTA oligomer, Asp306 and Asp309 rearrange (on loop8-9) as a consequence of the reconfiguration and shortening of loop10-9 on nucleic acid binding (*vide supra*, Fig. 4). Loop8-9 both lengthens and extends (towards α 10) and residues Asp306 and Asp309 move >10 Å from their position in the Apo structure to demask the positively charged pocket for dimerization. We argued that deletion of residues from loop10-9 may prevent the lengthening and extension of loop8-9 and the reconfiguration of residues Asp306 and Asp309 on nucleic acid binding. Indeed, when we delete four residues from loop10-9, SPARTA loses the ability to form dimers on nucleic acid binding (Supplementary Fig. 9), leading to almost complete loss of NADase activity (Fig. 5b).”

We fully agree with the reviewer that understanding how SPARTA becomes activated is a key question in the study of SPARTA – and we firmly believe that our manuscript is very unique in that respect. Our structural studies (crystallographic and cryo-EM) lay the essential basis for further experiments, not just in our laboratory but in that of many others.

We thank the reviewer for the very helpful query/question, which has greatly helped to improve the quality of the manuscript.

Reviewer #2:

“The authors have added an experiment where different mutations were made to further test their model, which have corroborated and expanded the previous work. In the opinion of this reviewer the manuscript is ready for publication.”

Response:

We thank the reviewer.

REVIEWERS' COMMENTS

Reviewer #1 (Remarks to the Author):

Thank you to the authors for taking the time to address this reviewers concerns and elaborating on the mechanisms of SPARTA activation. It is the opinion of this reviewer that the manuscript is now ready for publication.